# How to miss data? Reinforcement learning for environments with high observation cost

**Mehmet Koseoglu** [* 1]    **Ayça Özçelikkale** [* 2]

## Abstract

We consider a reinforcement learning (RL) setting where there is a cost associated with making accurate observations. We propose a reward shaping framework and present a self-tuning RL agent that learns to adjust the accuracy of the samples. We consider two different scenarios: In the first scenario, the agent directly varies the accuracy level of each sample. In the second scenario, the agent decides to perfectly observe some samples and miss others. In contrast to the existing work that focuses on sample efficiency during training, our focus is on the behavior of the agent when the observation cost is an intrinsic part of the environment. Our results illustrate that the RL agent can successfully learn that not all samples are equally informative and choose to observe the ones that are most critical for the task at hand with high accuracy.

## 1. Introduction

Doing experiments and collecting samples are expensive in various application scenarios, for instance in the case of medical applications with expensive tests or experiments with chemical processes, where the high cost can be due to limited human expert labor or materials. In such scenarios, the decision-maker has to decide whether an experiment which may reveal potentially important information but has a high cost should be conducted. This decision is challenging since the usefulness of the information is not certain and depends on the environment as well as the current knowledge state of the decision-maker. Hence, it is desirable to have an autonomous decision-making procedure which learns to find the optimum trade-off between the informative-

---
[*]Equal contribution  [1]Dept. Computer Engineering, Hacettepe University, Turkey [2]Dept. Electrical Engineering, Uppsala University, Sweden. Correspondence to: Mehmet Koseoglu <mkoseoglu@cs.hacettepe.edu.tr>.

*Presented at the first Workshop on the Art of Learning with Missing Values (Artemiss) hosted by the $37^{th}$ International Conference on Machine Learning (ICML).* Copyright 2020 by the author(s).

ness of the samples and their cost. In this article, we address this problem using a policy-gradient based reinforcement learning (RL) with reward shaping and provide a self-tuning RL agent that can learn to adjust the accuracy level of its samples, hence deciding effectively which samples to miss and which samples to collect.

This setting is closely related to the missing values problem (Little & Rubin, 1986) where some samples in a series of data are not available; hence, decision making have to be performed with an incomplete set of data. In particular, we consider the following question: "Which data samples are not informative and hence probably not harmful to miss or collect with high noise?". Not all values are equally informative, and there is a hierarchy of sample values in terms of their potential significance. The answer to this question reveals this hierarchy, which in turn will have prominent consequences for the processing of missing values.

The values that are missing but were expected to be uninformative can be processed in a manner that is different from the missing values that were expected to be informative. In particular, the values in the latter class need to be processed with more care since these values are expected to provide novel, distinguishing information. For instance, the practice of imputing with the mean of the variable (Little & Rubin, 1986) which induces a bias in the estimation procedure may be appropriate for the former case but not the latter.

**Contributions:** We propose a framework where a decision-maker can adjust the accuracy of its observations. We aim to decrease the accuracy of the observations as much as possible, and in the limiting case, only collect a small number of informative samples. In contrast to the existing work that focuses on sample efficiency during training, we consider the scenario where the observation cost is intrinsic to the environment. Using reward shaping, we provide a self-tuning RL agent that learns to successfully adjust the accuracy of the samples.

## 2. Related Work

A similar setting is active learning (Settles, 2010; Donmez et al., 2010) where an agent decides which queries to perform, i.e., which samples to take, during training. For in-

stance, an agent that learns to classify images can decide which images from a large dataset it would like to have labels for in order to have improved accuracy (Settles, 2010; Donmez et al., 2010). In a standard active learning approach (Settles, 2010) as well as its extensions in RL (Lopes et al., 2009), the main aim is to reduce the size of the training set, hence the agent tries to determine informative queries during training so that the performance during the test phase is optimal. In the test phase, the agent cannot ask any questions; instead, it will answer questions, for instance, it will be given images to label. In contrast, in our setting the agent continues to perform queries during the test phase, such as in the case of collecting camera images for autonomous driving. Our aim is to reduce the number of queries the agent performs during this actual operation.

Another related line of work consists of the RL approaches that facilitate efficient exploration of state space, such as curiosity-driven RL (Pathak et al., 2017) or active-inference based methods utilizing free-energy (Ueltzhöffer, 2018; Schwöbel et al., 2018) and the works that focus on operation with limited data using a model (Chua et al., 2018; Deisenroth & Rasmussen, 2011). In these works, the focus is either finding informative samples (Pathak et al., 2017) or using a limited number of samples/trials as possible (Chua et al., 2018; Deisenroth & Rasmussen, 2011) during the agent's training. In contrast to these approaches, we would like to decrease the number of samples taken during the test phase, i.e. actual operation of the agent.

## 3. Problem Setting

### 3.1. Preliminaries

Consider a Markov decision process (MDP) given by $\langle \mathcal{S}, \mathcal{A}, \mathcal{P}, R, P_{s_0}, \gamma \rangle$ where $\mathcal{S}$ is the state space, $\mathcal{A}$ is the set of actions, $\mathcal{P} : \mathcal{S} \times \mathcal{A} \times \mathcal{S} \to \mathbb{R}$ denotes the transition probabilities, $R : \mathcal{S} \times \mathcal{A} \to \mathbb{R}$ denotes the reward function, $P_{s_0} : \mathcal{S} \to \mathbb{R}$ denotes the probability distribution over the initial state and $\gamma \in (0, 1]$ is the discount factor.

The agent, i.e. the decision maker, observes the state of the system $s_t$ at time $t$ and decides on its action $a_t$ based on its policy $\pi(s, a)$. Here, the policy mapping of the agent $\pi(s, a) : \mathcal{S} \times \mathcal{A} \to [0, 1]$ is stochastic and gives the probability of taking the action $a$ at a state $s$. After the agent implements the action $a_t$, it receives a reward $r(s_t, a_t)$ and the environment moves to the next state $s_{t+1}$ which depends on the action $a_t$ and the previous state $s_t$. The aim of the agent is to learn an optimal policy mapping $\pi(s, a)$ so that the expected return $J(\pi) = \mathbb{E}_{a_t \sim \pi, s_t \sim P}[\sum_t \gamma^t r(s_t, a_t)]$ is maximized.

We adopt a deep RL approach, combining reinforcement learning with deep learning (François-Lavet et al., 2018). In particular, we consider a policy-based approach, Trust Region Policy Optimization (TRPO) (Schulman et al., 2015;

Hill et al., 2018). Here, the policy $\pi$ is directly characterized using a set of parameters $\eta$, i.e., $\pi(s, a) = \pi(s, a; \eta)$ and TRPO uses Kullbeck-Leiber divergence between the new policy and the old policy to create a trust region to improve convergence behaviour to the optimal policy. In our deep RL framework, $\eta$ represent the parameters of the multi-layer neural network that characterizes $\pi$.

### 3.2. Partial Observability

Although most RL algorithms are typically expressed in terms of MDPs, in most real-life applications the states are not directly observable. Hence, the data used by the agent to make decisions is not a direct representation of the state of the world, such as noisy images obtained from cameras or inaccurate test results in medical applications. Hence, we consider a partially observable Markov decision process (POMDP) where the above MDP is augmented by $\mathcal{O}$ and $\mathcal{P}_o$ where $\mathcal{O}$ represents the set of observations and $\mathcal{P}_o : \mathcal{S} \to \mathcal{O}$ represents the observation probabilities. Accordingly, the policy mapping is now expressed as $\pi(o, a) : \mathcal{O} \times \mathcal{A} \to [0, 1]$. In particular, the observations are governed by

$$o_t \sim p_o(o_t | s_t; \beta) \tag{1}$$

where $p_o(o_t | s_t; \beta)$ denotes the conditional probability distribution function (pdf) of $o_t$ under $s_t$ and is parametrized by $\beta$. Here, $\beta$ represents the accuracy of the observations. As $\beta$ increases, the accuracy of the observations decreases. For instance, consider the Gaussian additive noise model with $o_t = s_t + v_t$ where $o_t, s_t, v_t \in \mathbb{R}$ with $v_t$ Gaussian with $\mathcal{N}(0, \sigma_v^2)$. Then, one may choose $\beta = \sigma_v$ and hence parametrize $p_o$ as $p_o(o_t | s_t; \beta) = \mathcal{N}(s_t, \beta^2 = \sigma_v^2)$. Note that there is not a single choice for this parametrization, for instance, one may also adopt $\beta = \sigma_v^2$.

### 3.3. Proposed Approach

Reward shaping is a popular approach to direct RL agents towards a desired goal. Here, we would like the agent not only move towards to the original goal (which is encouraged by the original reward $r$), we also want it to learn to control $\beta$ so that not all samples are taken with the same accuracy. Hence, we propose reward shaping in the following form:

$$\bar{r} = f(r, \beta) \tag{2}$$

where $r$ is the old reward, $\bar{r}$ is the new reward and $f(r, \beta)$ is a monotonically increasing function of $r$ and $\beta$. Hence, the agent not only tries to maximize average of the original reward but it also tries to maximize the "inaccuracy" of the measurements. This can be equivalently interpreted as minimizing the cost due to accurate measurements.

In general, the form of the original reward $r$ depends on the particular application. Similarly, the form of the function $f(.)$ will be application specific. Nevertheless, a simple but

*Table 1.* Hyperparameters of the TRPO algorithm.

| PARAMETER | VALUE |
|---|---|
| COMPUTE GRADIENT DAMPENING FACTOR | 2.35E-05 |
| WEIGHT FOR THE ENTROPY LOSS | 0.01118 |
| GAMMA | 0.98 |
| GAE FACTOR | 0.9 |
| KULLBACK-LEIBLER LOSS THRESHOLD | 0.000193 |
| NO. OF TIMESTEPS TO RUN PER BATCH | 1024 |
| VALUE FUNC.'S NO. ITERS. FOR LEARNING | 10 |
| VALUE FUNC. STEPSIZE | 0.00428 |

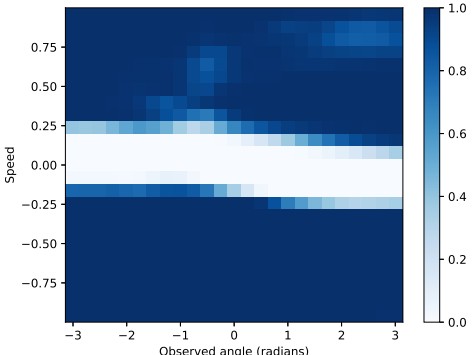

*Figure 2.* The noise level selected by the RL agent as a function of speed and angle of the pendulum.

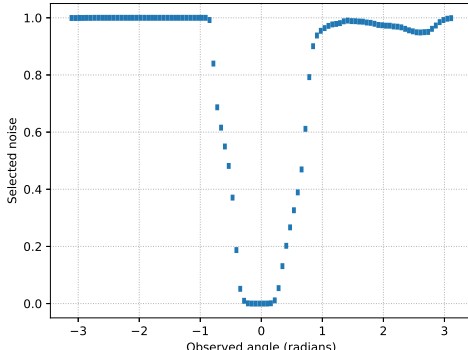

*Figure 1.* The noise level selected by the RL agent with respect to angle of the pendulum.

attractive option is the additive model

$$f(r, \beta) = r + g(\beta) \tag{3}$$

where $g(\beta)$ is an increasing function of $\beta$. Now, the design variable is the functional form of $g(.)$. In this article, we provide a proof of concept using (3). The details are provided in Section 4.

# 4. Experiments
## 4.1. Proposed Approach and Pendulum Environment

In our experiments, we consider Pendulum-v0 environment of the OpenAI Gym framework (Brockman et al., 2016). In the original pendulum environment, the state is defined by the tuple $(\theta, \dot{\theta})$ where $\theta \in (-\pi, \pi)$ is the angle of pendulum and $\dot{\theta} \in (-8, 8)$ is the angular speed of the pendulum. The observation is defined as $(\cos(\theta), \sin(\theta), \dot{\theta})$. The action of the agent is the effort exerted on the pendulum which is denoted as $\alpha \in (-2, 2)$. The pendulum starts at a random position. The aim is to bring the pendulum on the vertical position (i.e. $\theta = 0$) and keep it there.

In our modified environment, the noisy observations from the environment are given as follows:

$$\tilde{\theta} = \theta + C_\theta \times \mathcal{U}(-\beta, \beta), \tag{4}$$
$$\tilde{\dot{\theta}} = \dot{\theta} + C_{\dot{\theta}} \times \mathcal{U}(-\beta, \beta), \tag{5}$$

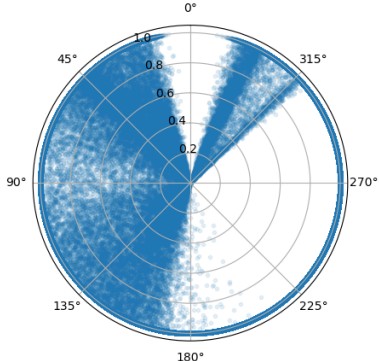

*Figure 3.* The noise levels selected by the RL agent as a function of the position of the pendulum.

where $\mathcal{U}(-\beta, \beta)$ is the uniform distribution over $[-\beta, \beta]$ with $\beta \in [0, 1]$. Here, $C_\theta$ and $\dot{C}_{\dot{\theta}}$ indicate the scaling of the noise level. In addition to the standard action of choosing $\alpha$, our agent also chooses $\beta$. The original reward function of the environment is given by (Brockman et al., 2016)

$$r = -(\theta^2 + 0.1\dot{\theta}^2 + 0.001\alpha^2). \tag{6}$$

We shape the reward according to the additive model

$$\bar{r} = r + g(\beta), \tag{7}$$

where $g(\beta) = k \times \beta$ and $k > 0$. As we reward the agent for acquiring noisy samples in proportion to $k$, the agent is expected to request more noisy samples as $k$ increases.

We consider the following scenarios:
*Scenario A:* The agent can vary the noise level continuously. We consider this scenario with $\beta \in [0, 1]$ and $C_\theta = 0.2 \times 2\pi$, $C_{\dot{\theta}} = 0.2 \times 8$ and $k = 1$. Hence, the maximum noise is 20% of the range of the respective noiseless observations.
*Scenario B:* The agent has to choose between i) collecting the observation with zero noise or ii) not getting it at all.

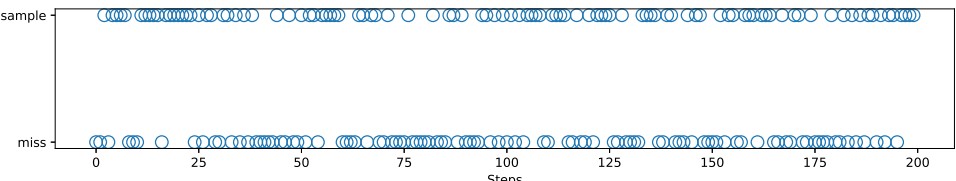

*Figure 4.* Sampling decisions made at each step of a sample run of the trained policy. In this run, the agent took 101 samples out of 200.

This setting corresponds to the case with $\beta \in \{0, \infty\}$. Here, missing values are imputed using the most recent sample.

For both scenarios, we inherited the hyperparameters of TRPO from (Raffin, 2018). The hyperparameters are presented in Table 1.

### 4.2. Experiments - Scenario A

Fig. 1 shows how the noise level selected by the RL agent changes with the angle of the pendulum. The agent reduces the noise level significantly when the observed position of the pendulum is close to the upright position since, in this vulnerable position, a badly chosen action based on a noisy measurement may cause the pendulum to drop.

In Fig. 2, we present the noise levels selected by the agent as a function of both speed and angle. These results reveal that the speed of the pendulum rather than its angle is the deciding factor for the sampling noise. If the speed of the pendulum is low, the agent gets a low-noise sample so that it can keep or move the pendulum around its upright position. Dependency of the noise on the angle in Fig. 1 is due to the fact that the speed of the pendulum is usually lowest in the upright position.

The noise levels of the samples for different positions of the pendulum are provided in Fig. 3. Each point corresponds to a sample, where the angular position of the point is the position of the pendulum and the distance of the point from the origin indicates the noise level. When the pendulum is in upright position, the agent gets samples without noise. We observe that the policy has an asymmetry in choosing the noisy samples in the angular domain. Instead of distributing the noiseless samples uniformly over the whole range of $\theta$ values, the agent preferred to always sample with the highest noise level when the pendulum is on the right side, between 180 and 310 degrees. This behaviour can be considered a consequence of the agent's behaviour for climbing up the pendulum in one side and letting it to fall on the other side.

### 4.3. Experiments - Scenario B

In Fig. 4, we provide the sampling decisions made by the agent when it is forced to decide between only sample and miss. Here, the agent only took half of the samples (101 out of 200) but was still able to exert correct actions and hold the pendulum in the upright position.

As a result of sparse sampling, it takes 47 steps to get to the upright position whereas it takes 43 steps when the agent has access to all noiseless samples. Hence, the number of samples required in order to reach the upright position is only slightly affected by the fact that the agent is penalized for taking samples. Note that the above 200 step window of Fig. 4, also shows what happens after reaching the goal position. As seen in the figure, the agent chooses to miss samples both before and after reaching the goal.

## 5. Discussions

These types of design problems are encountered in various applications, such as the case of expensive but high-accuracy medical tests, physics experiments as well as in wireless communications. For instance, consider the task of control over a wireless communication channel. Increasing the broadcast power, hence decreasing the effective noise of the communication channel, increases the information rate (Cover & Thomas, 1991). On the other hand, in practice, it is not possible to constantly use high amounts of broadcast power due to high energy footprint, health concerns, and also challenges due to high-power electronics. Hence, minimizing the communication power cost (i.e. controlling the inaccuracy of the measurements) while being able to perform the original task, such as the control of a pendulum, is highly desirable. These types of problems are typically approached by handling the communication and control tasks separately whereas, here, we provide a joint framework using an RL agent.

## 6. Conclusions and Future Work

Motivated by the varying cost of real-life experiments and their capacity to provide new information, we have proposed a framework where a decision maker can adjust the accuracy of its observations. We have considered an autonomous decision making setting where a RL agent learns to vary the accuracy of its samples using past experience. Generalizations to other observation accuracy models together with further investigations into optimal handling of missing and inaccurate samples are considered as interesting future research directions.

## Acknowledgements

This work is partially financially supported by Swedish Research Council under grant 2015-04011.

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
