# OpenReview forum: "How to miss data? Reinforcement learning for environments with high observation cost"
_ICML.cc/2020/Workshop/Artemiss — ICML Artemiss 2020_

### Official Review · AnonReviewer1 · 2020-06-22
**Interesting problem setting**

**Rating:** 6
**Confidence:** 4

**Review:**

This paper studies the interesting problem of RL when observations have a cost. The authors propose an approach that tunes the noise level of the observations to minimize cost while still solving the original RL problem.

The paper could be improved by providing some instantiations of real-life problems in the proposed framework (I can rather see these instantiations in the case in which observations are either not made or noise free than in the case where the noise level is controlled) and by providing reward information (how well are solutions found by proposed approach compared to solutions of the original RL problem and how much cost is saved by making noisy observations). Also, as there is some space in the paper left, it would be beneficial to spell out TRPO settings and the configurations of the used policy networks. The performed experiments are very simplistic and it is unclear how well the approach scales to more challenging environments.

I see the value of this paper mainly in the studied problem setting and not so much in the proposed approach which is not compared to other ad-hoc solutions for the problem setting. I appreciate the plots indicating that the learned policies tune the noise actively in an intuitive way.

---

### Decision · Program_Chairs · 2020-07-02

**Decision:**

Accept

**Comment:**

We're happy to accept this paper at Artemiss. We'll contact you soon to inform you about more details concerning the format of your presentation at the workshop, and the camera-ready version deadline. Please take into account the referee's comments to write the camera-ready version.